# Comparison of Quantification Methods to Estimate Farm-Level Usage of Antimicrobials Other than in Medicated Feed in Dairy Farms from Québec, Canada

**DOI:** 10.3390/microorganisms9051106

**Published:** 2021-05-20

**Authors:** Hélène Lardé, David Francoz, Jean-Philippe Roy, Jonathan Massé, Marie Archambault, Marie-Ève Paradis, Simon Dufour

**Affiliations:** 1Department of Pathology and Microbiology, Faculty of Veterinary Medicine, Université de Montréal, 3200 Rue Sicotte, Saint-Hyacinthe, QC J2S 2M2, Canada; jonathan.masse.1@umontreal.ca (J.M.); marie.archambault@umontreal.ca (M.A.); 2Regroupement FRQNT Op+lait, 3200 Rue Sicotte, Saint-Hyacinthe, QC J2S 2M2, Canada; david.francoz@umontreal.ca (D.F.); jean-philippe.roy@umontreal.ca (J.-P.R.); 3Groupe de Recherche en Épidémiologie des Zoonoses et Santé Publique, Faculty of Veterinary Medicine, Université de Montréal, 3200 Rue Sicotte, Saint-Hyacinthe, QC J2S 2M2, Canada; 4Department of Clinical Sciences, Faculty of Veterinary Medicine, Université de Montréal, 3200 Rue Sicotte, Saint-Hyacinthe, QC J2S 2M2, Canada; 5Association des Médecins Vétérinaires Praticiens du Québec, 1925 rue Girouard Ouest, Saint-Hyacinthe, QC J2S 3A5, Canada; marie-eve.paradis@amvpq.org

**Keywords:** farm-level, monitoring, antibiotic, antimicrobial, dairy cattle, veterinary invoice, animal health record, herd management software, quantification method, antimicrobial use

## Abstract

The objective of the study was to compare three quantification methods to a “garbage can audit” (reference method, REF) for monitoring antimicrobial usage (AMU) from products other than medicated feed over one year in 101 Québec dairy farms. Data were collected from veterinary invoices (VET method), from the “Amélioration de la Santé Animale au Québec” provincial program (GOV method), and from farm treatment records (FARM method). The AMU rate was reported in a number of Canadian Defined Course Doses for cattle (DCDbovCA) per 100 cow-years. Electronic veterinary sales data were obtained for all farms for VET and GOV methods. For the FARM method, a herd management software was used by 68% of producers whereas farm treatment records were handwritten for the others; records could not be retrieved in 4% of farms. Overall, agreement was almost perfect between REF and VET methods (concordance correlation coefficient (CCC) = 0.83), but moderate between REF and GOV (CCC = 0.44), and between REF and FARM (CCC = 0.51). Only a fair or slight agreement was obtained between any alternative method of quantification and REF for oral and intrauterine routes. The billing software used by most of Québec’s dairy veterinary practitioners seems promising in terms of surveillance and benchmarking of AMU in the province.

## 1. Introduction

Farm-level quantification of antimicrobial use (AMU) is essential to study and better understand relationships with antimicrobial resistance [1]. Finding a reliable way to quantify AMU is a prerequisite for implementation of surveillance and benchmarking programs [2] with reduction targets. The characteristics of a good AMU monitoring system would be: to be representative of the actual AMU (close to or well correlated to the real AMU), sustainable, computerized and centralized to facilitate data extraction and production of reports, with data easily collected and analyzed (programmable), and obtained from the highest number of participants [3,4]. In addition, the output of a monitoring system should be easily understood by the end-users.

Different quantification methods have been used to report AMU in producing animals. One of them involves the use of a specifically provided receptacle to collect empty medication containers (vials, bottles, packaging, syringes, etc.) on farms (garbage can audit) and is traditionally considered to be a reliable way to collect data [5]. This method was used in Canada to report AMU in beef [6] and dairy cattle [7,8]. It relies on the active participation of producers, the regular collect of garbage bags, and a manual inventory by the research team. This method was judged reliable to report AMU [9,10], but seemed to underestimate AMU for intramammary infusions [11], and oral and intrauterine boluses [8,12]. Moreover, obtaining sustainable data from a large proportion of farms seems unfeasible (too labor-intensive and time-consuming), hence, a surveillance program could hardly rely on this quantification method.

Alternative methods of quantification have been described in the literature, and some of them are now the foundation of monitoring systems for reporting farm-level AMU in various countries [13]. Veterinary data (invoices, prescriptions, or animal health records) were successfully used in Austria [14,15,16] and compiled higher quantities of antimicrobials than the garbage can audit system [17]. Collecting data directly from farm records (individual animal treatment records) could theoretically be used as a quantification method [18,19,20], especially in countries where completion of animal health records is mandatory. This method would rely on the thorough completion of records by producers and the decoding of treatment lines. In many settings, collecting these data is impaired by the use of paper vs. electronic treatment records. Missing data were often highlighted when quantities from farm records were compared to a garbage can audit [9,17]. Currently, we are not aware of any country-level surveillance program based only on farm treatment records.

In Canada, no monitoring system is currently in place for surveillance of AMU in dairy farms [13]. However in the province of Québec, several quantification methods could potentially be used for this purpose: (i) the veterinary sales data, (ii) the governmental database from the provincial program Amélioration de la Santé Animale au Québec (ASAQ) that includes eligible veterinary invoices produced by a veterinarian during a farm visit and submitted electronically to the Ministère de l’Agriculture, des Pêcheries et de l’Alimentation du Québec (MAPAQ) [21,22], and (iii) data collected from farm treatment records as part of the ProAction on-farm food safety program (mandatory in Québec since the late 1990s) [23]. These three sets of data could be used to quantify the AMU from products other than medicated feed. Although medically important antimicrobials were uncommonly used in medicated feed in dairy farms in Canada [8], there would still be a need for monitoring this usage as well. However, in this case of AMU, although a veterinary prescription is needed, antimicrobials are generally sold by feed mills. The three sets of data identified as potential ways to quantify AMU would not be able to quantify usage through animal feed.

The objectives of this study were, therefore, to compare available methods for quantifying AMU in products other than medicated feed in dairy farms from Québec, Canada, in order to identify a promising method to develop a surveillance and benchmarking system. We hypothesized that veterinary sales data would be correlated well with the real farm AMU (measured using a garbage can audit) and could be the basis for a provincial AMU monitoring system on dairies. We hypothesized that data from the governmental program or from farm treatment records would be correlated with the reference method to an acceptable level, but that the correlation would be lower.

This study is part of a larger study on AMU in Québec dairy farms.

## 2. Materials and Methods

This study was approved by the Health Research Ethics Committee of the Université de Montréal (project number 16-163-CERES-D). Throughout the paper, we followed the Strengthening the Reporting of Observational Studies in Epidemiology -Veterinary Extension (STROBE-Vet) statement guidelines [24,25] for reporting (Appendix A).

### 2.1. Recruitment of Participants

We recruited 101 dairy farms (including two organic farms) for one year (2017–2018) in the Canadian province of Québec. Briefly, dairy farms were randomly selected in three regions of the province (Centre-du-Québec, Estrie, and Montérégie). The recruitment represented the region to province ratio regarding number of farms: 19%, 15%, and 10% for Montérégie, Centre-du-Québec, and Estrie, respectively (stratified random sampling). Dairy farms were excluded if: (i) replacement animals were not raised on the farm or some animals were kept in a barn shared with cattle from another farm; (ii) a cessation of activity was planned in the coming 12 months; or (iii) the farm was already recruited in a pilot stage for the current project (i.e., for questionnaire development). The possibility of a selection bias was quantified, and a detailed description of the recruitment was provided by Lardé et al. [8]. Demographic information of the 101 farms is presented in Appendix A.

### 2.2. Quantification of Antimicrobial Use (AMU)

Four methods were identified for quantification of the AMU in Québec dairy farms, and characteristics of the data for each method (source, type, extraction method) with provincial specificities are described in Appendix B.

#### 2.2.1. Garbage Can Audit (Reference Method, REF)

Drug packaging was collected in a dedicated garbage can, which was placed on each farm (10 visits planed par farm over one year), and a thorough inventory of products containing antimicrobials was performed over one year [8]. This method was considered in our study as the reference method since it can be generalized to different countries and settings.

#### 2.2.2. Veterinary Invoices (VET Method)

Veterinarians in charge of the 101 farms all agreed to share invoices issued between the start and end date of the project (for each farm). Different types of billing systems were identified. For 100% of the farm, Vet-Expert software or Sysvet software were used for invoicing. In addition, for 13 farms, some invoices originated from a billing system other than Vet-Expert or Sysvet.

For each invoice, information collected was the invoice number and date, name of the farm, name of the veterinarian, name and code of the item sold, quantity and format of the item, invoiced amount, type of animals (bovine, equine, etc.), some information about the prescription (route of administration, treatment duration, dosage, frequency), and whether the invoice would be transferred to the governmental ASAQ program for reimbursement. In the data set each line corresponded to one product from one given invoice. A few lines with an amount to be paid equal to zero (suspected to correspond to products that were administered by a veterinarian from a bottle that was already owned by the producer and present on the farm) and lines explicitly dedicated to species other than bovine were removed. Only invoices matching the 12 months garbage can audit study were retained.

#### 2.2.3. Governmental Database from the Amélioration de la Santé Animale au Québec (ASAQ) Program (GOV Method)

Veterinary invoices that were eligible as part of the governmental ASAQ program were identified as such in veterinary billing systems, and, for simplicity, were extracted from the previously described database. We collected the same information from ASAQ invoices as the one obtained from invoices collected for the VET method. Again, only data matching the garbage can audit period were retained.

#### 2.2.4. Farm Treatment Records (FARM Method)

For each farm, a copy of all treatment records corresponding to the farm study period was collected during the last visit of the project. Two types of records were identified: computerized using a herd management software program (64 farms), handwritten (32 farms), or a combination of computerized and handwritten records (5 farms).

Paper records were read a first time by a member of the research team. Information stored for each line of treatment was: farm identification number, animal identification number, date of treatment, indication, product, dose, unit, and route of administration. Then, paper records were read a second time by another member of the research team, corrections were applied if errors were noted, and the most likely value was applied, when possible, for lines with illegible handwriting.

Data from all computerized treatment records, and from the transcript of paper records were merged into one Excel spreadsheet. If the dose was not recorded, the most commonly reported dose in the study was assigned to the line of treatment. If the type of product was not clearly defined, then the most likely and common form of the product in the study was used (“penicillin” was assigned to “procaine penicillin G 300,000 IU/mL” for example).

### 2.3. Statistical Analyses for Quantification of AMU

All datasets (REF, VET, GOV, and FARM) were imported and merged using SAS 9.4 statistical software for further data processing. For each farm and method, a quantity (in grams) was calculated by antimicrobial agent and route of administration for the study period (Equation (1)).
Quantity (grams) of antimicrobial A administered through route R = ∑ (number of units of product Pi (mL, tubes, boluses, etc.) × antimicrobial concentration in Pi (g/unit))(1)
A: antimicrobial agent; R: route of administration (intramammary for lactating cows, intramammary at dry-off, injectable, intrauterine, oral, or topical); Pi: product(s) containing A and administered through R.

The quantity (grams) of antimicrobial A was then converted into a number of Canadian Defined Course Doses for cattle (DCDbovCA; [26]) and standardized by 100 cows and by 365 days to report AMU rates in a number of DCDbovCA/100 cow-years (Equation (2)). The number of cows was obtained from an in-person interview with the 101 producers (total of milking and dry cows at the time of the interview) and assumed to be stable over the 12-month period.
AMU rate (number of DCDbovCA/100 cow-years) for antimicrobial A administered through route R = (quantity (grams) of antimicrobial A obtained from (Equation (1)) × 100 × 365) / (DCDbovCA value (g per animal per course) × number of cows in the farm x number of days in the study)(2)

Note that antimicrobials without a DCDbovCA value assigned (i.e., from products not labeled for cattle in Canada) were simply left on a gram unit basis. The AMU rates by antimicrobial agent and route of administration were then collapsed to present total AMU, AMU by route of administration (intramammary for lactating cows, intramammary at dry-off, injectable, oral, intrauterine, or topical), and AMU by category of importance in human medicine according to the World Health Organization (WHO) [27] and to Health Canada [28].

To estimate the mean AMU in the target population (Québec dairy farms), for each method of quantification and for total AMU, AMU by route of administration, and AMU by antimicrobial category according to the WHO or to Health Canada, we used a negative binomial regression model with the non-standardized number of DCDbovCA as outcome, the natural logarithm of the number of cow-years of follow up as offset, and without including any fixed predictors. Intercept estimates and their 95% confidence interval (95% CI) were then back-transformed and multiplied by 100 to report AMU rates in DCDbovCA/100 cow-years. If a problem of dispersion was identified (defined as a Pearson’s chi-square/degrees of freedom <0.80 or >1.20), then robust standard error was used to compute 95% CI [29].

To evaluate the strength of agreement between each alternative method and the REF method, Lin’s concordance correlation coefficients (CCC) were computed for the total AMU rate, by route of administration, and by category of importance in human medicine, using R statistical software (R version 4.0.4 for Windows, package ‘epiR’ version 2.0.19) [30,31]. We applied the scale developed by Landis and Koch [32] to interpret the observed CCC: almost perfect (0.81–1.00), substantial (0.61–0.80), moderate (0.41–0.60), fair (0.21–0.40), or slight agreement (0.00–0.20). Bland–Altman diagrams were plotted using R statistical software (package ‘ggplot2′ version 3.3.3). Mean differences (biases) and 95% limits of agreement were determined between the total AMU rate obtained from the REF method vs. each alternative method (package ‘epiR’ Version 2.0.19) [33,34]. Mean differences were also reported by route of administration and by antimicrobial category.

To further describe the relationship between alternative methods and REF, we used the negative binomial models previously described with the number of DCDbovCA (total AMU only) obtained from VET, GOV, and FARM methods (one at a time) as outcome and the total AMU rate obtained from the REF method as sole predictor. The fit of each model was verified using a Pearson chi-square test and adjusted using robust variance if needed. The linearity of each relationship was verified by visual inspection of a scatter plot with a loess curve. If a non-linear relationship was suspected visually, then polynomial terms were added to the model. Polynomial terms were considered statistically significant whenever the *t*-test for the coefficient of that term yielded a *P*-value ≤ 0.05 [35]. Predictions from the models were then used to estimate the total AMU rates that would be obtained with VET, GOV, and FARM methods, when a total usage rate of 0, 50, 100, 400, or 800 DCDbovCA/100 cow-years was identified with REF.

## 3. Results

### 3.1. Descriptive Data for the Different Methods of Quantification

Antimicrobial agents identified during the study by at least one of the quantification methods (REF, VET, GOV, or FARM) and quantified in DCDbovCA are presented in Appendix C with their route(s) of administration and categorization according to the WHO or to Health Canada. The 101 dairy producers recruited for the study bought veterinary products from 35 veterinary facilities. As billing systems, 24 veterinary facilities used only the Vet-Expert software throughout the project, four used only the Sysvet software, three changed from Sysvet to Vet-Expert software, and four used their own billing program. The four veterinary facilities that had their own billing system were not the primary veterinarians of the related farms of the project.

Most of the producers (84/101) bought products from only one veterinary facility for the entire project, 12 had invoices from two veterinary sources, and five from three sources. Among the producers that had several sources of invoices, 12 had invoices from the veterinary teaching hospital (VTH) of the faculty of veterinary medicine (Université de Montréal), because at least one animal was referred for evaluation and treatment during the timeframe of the project.

Thus, we extracted 4483 invoices from Vet-Expert software (87 farms had at least one invoice from Vet-Expert), 1196 invoices from Sysvet software (28 farms had at least one invoice from Sysvet), 26 invoices from VetView software of the VTH (12 farms), and 35 invoices from another billing program (3 farms).

We obtained 22,616 sales observations (lines) from the combined veterinary sales data. After removal of lines with an amount to be paid equal to zero (565 lines, 2.5% of lines) and of lines specifically mentioned to be dedicated to canine, feline, or equine species (102 lines, 0.5% of lines), a total of 21,949 lines remained for analysis for the VET method. Similarly, 16,244 observations were eligible and submitted for the governmental ASAQ program. After removal of lines with an amount to be paid equal to zero (407 lines, 2.5% of lines) and of lines specifically mentioned to be dedicated to canine, feline, or equine species (75 lines, 0.5% of lines), a total of 15,762 lines remained for analysis for the GOV method.

Among herd management software used to compile animal treatments, DSA Laitier-Producteur (*n* = 48; DSAHR Inc., Saint-Hyacinthe, QC, Canada) and Lac-T (*n* = 17; LactoLogic Inc., Drummondville, QC, Canada) were mainly represented, followed by T4C (*n* = 6; Lely, Maassluis, The Netherlands), DelProTM (*n* = 2; DeLaval International AB, Tumba, Sweden), HerdMetrix (*n* = 2; UNIFORM-Agri, Assen, The Netherlands), and DairyPlan C21 (*n* = 2; GEA, Düsseldorf, Germany). Note that eight producers used two types of software. Four producers did not provide any treatment records at the end of the year project. They reported using a paper register but were not able to present it during the last visit nor after two reminders by phone call after the last visit.

### 3.2. Quantification of the AMU by the Four Methods

Distributions of the total AMU rate for the 101 (REF, VET, and GOV methods) or 97 (FARM method) dairy farms are presented in Figure 1 by quantification method. We estimated a total usage rate of 416 (95% CI: 362, 477), 520 (95% CI: 455, 595), 229 (95% CI: 189, 278), and 275 (95% CI: 232, 328) DCDbovCA/100 cow-years from REF, VET, GOV, and FARM methods, respectively. The total AMU rate was thus significantly lower for GOV and FARM methods, but not statistically different for the VET method, when compared to the rate estimated by the REF method, indicating that the VET method was more reliable than GOV and FARM methods for monitoring the total AMU in Québec dairy farms.

Percentages of dairy producers using antimicrobials and estimates of the usage rate by route of administration and according to the categories of WHO and Health Canada are presented for the four methods of quantification in Table 1. All estimates of the usage rate according to VET were numerically greater than estimates from REF. Estimates according to VET were statistically different from those of the REF method for injectable, oral, and intrauterine formulations, as well as for highly important antimicrobials and antimicrobials not used in humans (according to WHO categorization), and, finally, for Health Canada’s Category III and Category IV antimicrobials.

Estimates of the usage rate according to GOV were all numerically lower than estimates from REF except for oral, intrauterine, and Health Canada’s Category III antimicrobials where they were higher. All GOV and REF estimates were statistically different, with the exception of injectable, oral, and Health Canada’s Category III antimicrobials. Only for these exceptions, the GOV method showed AMU rates comparable to the REF method.

Finally, estimates of the usage rate according to FARM were all numerically lower than estimates from REF except for oral and intrauterine routes where they were higher. All FARM and REF estimates were statistically different, with the exception of dry cow intramammary and oral routes, antimicrobials not used in humans (WHO categorization) and Health Canada’s Categories III and IV antimicrobials. Only for these exceptions, the FARM method showed AMU rates comparable to the REF method.

### 3.3. Comparisons between Methods of Quantification

Concordance plots (CCC plots) and Bland–Altman plots between the REF method and each alternative method (VET, GOV, and FARM) are presented for the total AMU rate in Figure 2. An almost perfect agreement was observed between VET and REF methods (Figure 2A; CCC = 0.83; 95% CI: 0.76, 0.87). A moderate agreement was observed between GOV and REF methods (Figure 2C; CCC = 0.44; 95% CI: 0.31, 0.55) and between FARM and REF methods (Figure 2E; CCC = 0.51; 95% CI: 0.38, 0.63). The Bland–Altman plots did not highlight any obvious patterns of disagreement between methods. The farm-level mean difference between VET and REF methods was +105 DCDbovCA/100 cow-years (Figure 2B; 95% CI: −237, 446). On the other hand, using the governmental database (GOV method) or the farm treatment records (FARM method), farm-level mean differences compared to the REF method were −187 DCDbovCA/100 cow-years (Figure 2D; 95% CI: −673, 300) and −145 DCDbovCA/100 cow-years (Figure 2F; 95% CI: −635, 344), respectively. Thus, the farm-level mean differences were not statistically different between the REF method and each alternative method.

Values of the CCC (95% CI) and mean difference (95% CI) are presented by route of administration and by category according to WHO and Health Canada in Table 2. An almost perfect agreement was identified between VET and REF methods for almost all routes and categories, except for the WHO’s highly important antimicrobials (moderate agreement), Health Canada’s Category III antimicrobials (slight agreement), oral (slight agreement) and intrauterine (fair agreement) routes. Health Canada’s Category III belongs to phenicols (injectable products containing florfenicol), tetracyclines (injectable, intrauterine, and oral products containing oxytetracycline, oral products containing tetracycline), and sulfonamides (oral and intrauterine products). Substantial agreement was identified between GOV and REF methods only for the injectable route, and between FARM and REF methods for the intramammary route (drying-off), the injectable route, and Health Canada’s Category I antimicrobials. For other routes of administration and categories, a moderate or poorer agreement was identified.

Relationship between a given AMU rate estimated using the REF method and what would be observed using VET, GOV, or FARM methods is presented in Table 3 to illustrate the relationship between quantification methods. The relationship between the total AMU rate according to each alternative method and REF was visually linear, and thus, no polynomial terms were added in the models.

Products containing antimicrobials that could not be quantified in DCDbovCA were rarely identified by the four methods of quantification and are described in Appendix A (compounded forms, veterinary products labeled for species different from cattle or no more labeled for cattle, and human products).

## 4. Discussion

The VET method seemed really well suited for monitoring of farm-level AMU in Québec dairy farms. The billing software program Vet-Expert, used with this method, was employed by a majority of veterinary practitioners in our study, and was even more popular in 2021 (>90% of large animal veterinarians in the province of Québec; personal communication, M.-È.P.). The software was maintained by the company DSAHR, a subsidiary of the Association des médecins vétérinaires praticiens du Québec (AMVPQ) and was used for dairy and beef cattle, small ruminants, and equine species. Thus, surveillance could possibly be enlarged to those species. Data were centralized and easily extracted by the DSAHR software’s programmers, with the approval of producers and veterinarians. Programming could be standardized to generate benchmarking reports on AMU at different scales: at farm-level, veterinary-level (or veterinary facility-level), region-level, or province-level.

In Canada, a veterinary prescription is mandatory for a producer to legally buy a veterinary product, and thus a majority of veterinary products are prescribed, but also sold by the veterinarians to the producers [21]. However, one limitation when reporting AMU using prescriptions or invoices is the uncertainty that the product bought would be used without delay, entirely, for the targeted species (not always indicated in invoices) or as prescribed [36]. In Canada, most of the veterinary products are currently sold by bottle or complete packaging. The accuracy of the estimated AMU rate would be increased if a precise dose, calculated for a given animal, would be sold (the exact amount of product needed to treat a given animal). In our methodology, using veterinary prescriptions instead of invoices would have increased the risk of overestimation of the actual usage (all prescribed products are not necessarily bought).

A slight (oral route) and fair (intrauterine route) agreement was found between VET and REF methods. This finding may be due to the poor ability to collect individual units for oral and intrauterine formulations with the garbage can audit. This was corroborated by a recent Swedish study [12], especially for intrauterine drugs. The authors hypothesized that the drug packaging was routinely discarded by the veterinarian after treatment within the disposable sleeve, but did not reach the garbage can, and we believe that the same is happening in Canada. Furthermore, the majority of oral boluses were sold in boxes of several units, and these boxes were rarely collected from the garbage can. We can hypothesize that (i) veterinarians sometimes sold the bolus per unit, without the packaging (hence, there is nothing to throw in the garbage can after administration to the animal), or (ii) producers could have forgotten to throw the empty box in the garbage can for those specific products or (iii) the empty box could have been put in the recycle bin instead of the dedicated garbage can of the project. The VET method was efficient in identification of oral and intrauterine units (present in veterinary invoices even if only one unit was sold from a several unit packaging).

In our study, the GOV method offered the advantage of being governmental and was originally considered as the best method to oversee the implementation of a monitoring system. However, the governmental database was not judged suitable for any level of AMU surveillance in Québec dairy farms (in general, but also by route of administration or by category). The database from the GOV method was a subset of the one from the VET method, therefore it was expected that the GOV method would not perform as well as the VET method. Interestingly, we did not use the database from the MAPAQ directly: our database for the GOV method was generated using veterinary invoices identified as eligible, and thus submitted for the governmental ASAQ program. Some of these invoices could further be discarded from the governmental database. Thus, the true governmental database may even yield lower AMU estimates than those obtained. Nevertheless, the database used as a proxy for the GOV database did not covered enough sales of antimicrobials to correlate adequately with the REF method.

Treatment records (FARM method) performed to an intermediate extent compared to VET and GOV methods. In particular, a substantial agreement was found between FARM and REF methods for Health Canada’s Category I, meaning that products containing third generation cephalosporins, polymyxins, or fluoroquinolones could be monitored through analysis of treatment records. The agreement was moderate for WHO’s highest priority critically important antimicrobials, the only difference of this category in comparison with Health Canada’s Category I being macrolides [27,28]. Macrolides are not labeled for lactating cows in Canada, and we suspected that treatments for non-lactating animals were not always written in the treatment records, as compared to treatments for lactating cows (which have milk withdrawal). Even if the FARM method performed correctly for Health Canada’s Category I, collecting data was difficult (no standardization, handwriting, incomplete information) and time-consuming with the problem of underreported AMU as observed by other authors [37,38]. Advantages of the FARM method were that results could be generalized to other Canadian provinces (the ProAction program being pan-Canadian) and that we were confident that products were actually used to treat dairy animals.

We can hypothesize that our results could be generalized to the Québec province because we recruited 101 farms in three central dairy regions with a good veterinary service coverage (Montérégie, Centre-du-Québec, Estrie). Results could have been different in remote regions where veterinary services are sparse, with a small number of veterinarians per facility, or with smaller farms. Some authors showed a positive association between herd size and the AMU rates [8,39] meaning these rates were observed to be higher in larger herds. However, it is expected that the VET method would perform as well in the entire Québec province as in our sample, because Vet-Expert software is used uniformly in the province. Concerning the FARM method, the filling in of treatment records could vary in regions where access to veterinary care is limited. The 101 producers interviewed as part of this project believed that they recorded a mean of 91% (range from 20% to 100%, median 100%) of antimicrobial treatments in their farm records, in comparison with 84% (range from 0 to 100%, median 95%) of non-antimicrobial treatments. We believe that they overestimated the real proportion of written treatments, based on the FARM method performances.

According to a recent review on existing systems for AMU monitoring at the farm-level [4], as of March 2020, 13 active systems from 11 European countries were identified for dairy cattle. Input of AMU data for these 13 systems was mostly from veterinarians (11/13) but could be provided (completely or partially) by farmers (2/13), feed mills (2/13), pharmacies (3/13), researchers (1/13), or not specified (1/13). Participation of the farms to the monitoring system was compulsory by legislation in most of the cases (6/13). Only one system was characterized by a voluntary participation (DLN cattle in Czech Republic). Benchmarking and reporting at farm-level was part of the monitoring system in 8/13 cases. According to Sanders et al. [4], three general features need to be evaluated when developing a monitoring system: (i) the coverage (proportion of the animal population included from the animal sector targeted by the system), (ii) the main funder (private or governmental), and (iii) the participation in the system (voluntary or compulsory). Using the software Vet-Expert specifically to build a monitoring system of the AMU at the farm-level in the Québec province would present the following features: (i) coverage of almost 86% (2018) to 90% (2020) of Québec dairy animals; (ii) funding (maintenance) of Vet-Expert is private, the company being owned equally by the AMVPQ and the Centre de Distribution de Médicaments Vétérinaires (CDMV), a Canadian distributor of veterinary products and services. However, development of a monitoring system through this software could be part of a collaboration and use governmental funding, and the government could, in return, commission benchmarking reports; (iii) participation in the system is expected to be voluntary (in Canada, invoices, once issued, belong to the producer, thus consent of producers should be obtained for farm-level reports). In addition to these characteristics, the fact that the data are centralized would facilitate data extraction.

## 5. Conclusions

Quantification and monitoring of AMU at the farm-level is an essential step in the fight against antimicrobial resistance [40,41]. For the Québec province in Canada, several quantification methods were identified as candidates for becoming a monitoring system of AMU for dairy cattle. Invoices submitted to the governmental ASAQ program did not provide sufficient information to estimate adequately AMU, and this method was not well correlated with the REF method. Moreover, the ASAQ program is expected to mutate in coming years, and the database could no longer be available. Farm records provided precise data (exact doses per animal per treatment) on the real usage of antimicrobials for treatment of dairy animals on the farm, but they were suspected to be incomplete and were only moderately correlated with the reference method. These records were not standardized from one farm to another, and collection of data was difficult (missing data, handwritten or incomplete lines of treatment). The method using veterinary invoices for quantification of the AMU correlated almost perfectly with the reference method.

Using the database extracted from veterinary billing software is a promising method to quantify and monitor AMU in Québec dairy farms. The system has a nearly full coverage of the dairy farms in the province and would allow farmers or veterinarians to be benchmarked, which is comparable to other existing systems for AMU monitoring at the farm-level [4]. Additional standardization of data entry would ease and reduce errors in analyses. The development of an interactive interface available online for presenting results at different scales (farm, veterinary practitioner, veterinary facility, region, province) to end-users (dairy farmers and their veterinarians) would strengthen this emerging surveillance system. In the future, this system would help fight antimicrobial resistance by guiding the development of new policies.

## Figures and Tables

**Figure 1 microorganisms-09-01106-f001:**
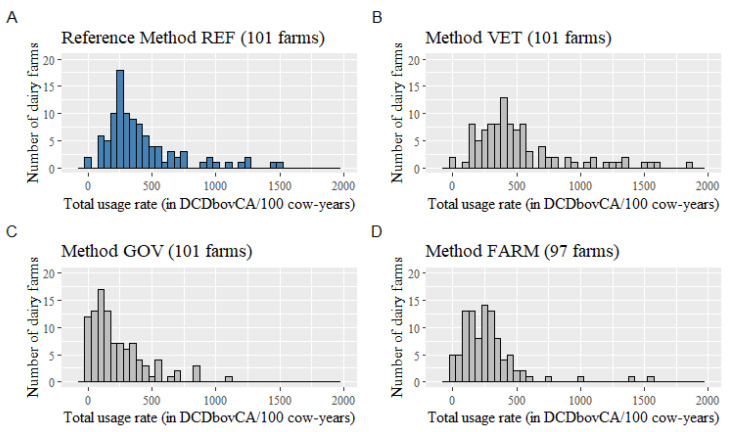
Distribution of the total antimicrobial usage rate (from products other than medicated feed) in Canadian Defined Course Doses for cattle (DCDbovCA)/100 cow-years, quantified using four different methods from 101 Québec dairy farms: (**A**) garbage can audit (REF method), (**B**) all veterinary invoices (VET method), (**C**) veterinary invoices eligible for the Québec governmental program (GOV method), and (**D**) farm treatment records (FARM method; note that farm treatment records could be retrieved from only 97 dairy farms).

**Figure 2 microorganisms-09-01106-f002:**
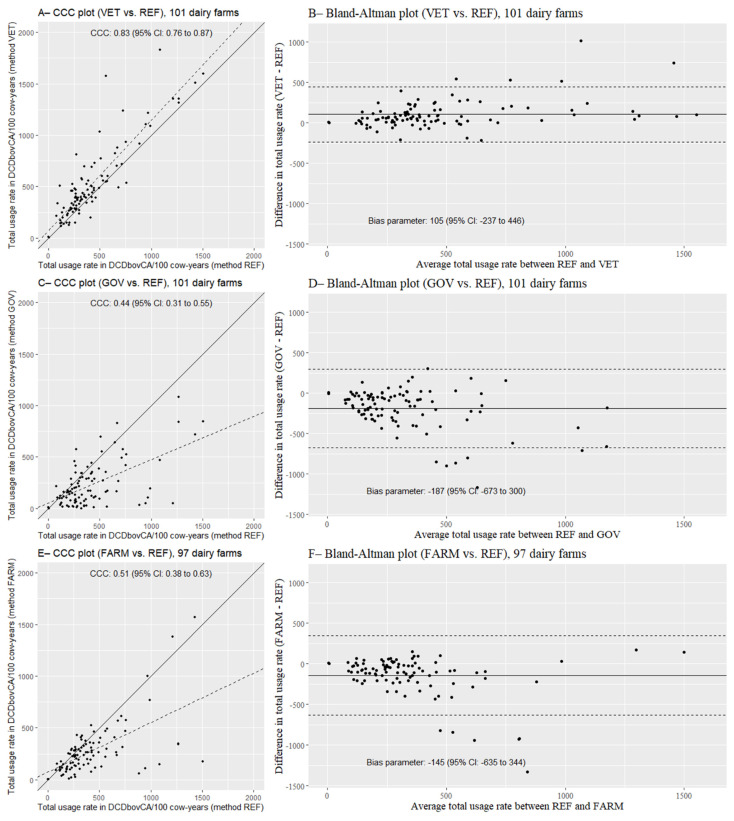
Concordance plots (concordance correlation coefficient (CCC) plots) and Bland–Altman plots showing agreement in quantification of the total usage rate of antimicrobial agents in products other than medicated feed (in DCDbovCA/100 cow-years) between a garbage can audit (REF method) and alternative methods using veterinary invoices (VET method; **A**,**B**), a governmental database (GOV method; **C**,**D**), and farm treatment records (FARM method; **E**,**F**). Each black point represents one farm of the project (101 farms in **A**–**D**, 97 farms in **E**,**F**). In CCC plots, the solid and dashed lines represent, respectively, the line of perfect concordance and the reduced major axis; concordance correlation coefficients (CCC) are presented with their 95% confidence interval (95% CI). In Bland–Altman plots, the solid line and the dashed lines represent, respectively, the mean difference (estimated bias) and the 95% limits of agreement; the mean difference between the REF method and the alternative method is presented with its 95% confidence interval (95% CI).

**Table 1 microorganisms-09-01106-t001:** Percentage of dairy producers using antimicrobial agents in products other than medicated feed, and estimates of the usage rate in Canadian Defined Course Doses for cattle (DCDbovCA)/100 cow-years (95% confidence interval, 95% CI) by route of administration, by category according to the World Health Organization (WHO) or to Health Canada, and in general, estimated using negative binomial regression models applied to four quantification methods in 101 or 97 dairy farms from Québec, Canada: garbage can audit (REF method), all veterinary invoices (VET method), veterinary invoices eligible for the Québec governmental program (GOV method), and farm treatment records (FARM method).

Antimicrobial Category		Reference Method REF (101 Farms)		Method VET (101 Farms)		Method GOV (101 Farms)		Method FARM (97 Farms)	
		% of Users	Estimated Rate (95% CI)	% of Users	Estimated Rate (95% CI)	% of Users	Estimated Rate (95% CI)	% of Users	Estimated Rate (95% CI)
Route of administration	Intramammary for lactating cows	97	257 (210, 315)	95	275 (220, 344)	77	118 (89, 155)	92	137 (105, 178)
	Intramammary at dry-off	95	73 (65, 82)	93	73 (65, 83)	54	25 (19, 34)	94	65 (57, 75)
	Injectable	98	64 (54,76)	100	77 (65, 92)	99	46 (39, 55)	97	38 (29, 49)
	Oral ^1^	36	19 (13, 27)	82	78 (58, 105)	64	29 (20, 42)	56	27 (20, 37)
	Intrauterine	29	3 (2, 5)	86	17 (12, 23)	83	13 (10, 17)	61	9 (6, 13)
	Topical	0	-	4	1 (0, 2)	3	0 (0, 1)	0	-
WHO ^2^	Highest Priority Critically important	97	94 (77, 113)	99	101 (83, 122)	90	44 (35, 57)	96	53 (40, 69)
	High Priority Critically Important	82	55 (44, 70)	75	61 (48, 78)	58	25 (19, 34)	75	28 (21, 39)
	Highly important	99	178 (158, 200)	100	245 (216, 278)	99	112 (94, 133)	100	124 (107, 143)
	Not used in humans	92	89 (76, 105)	94	113 (99, 130)	73	48 (39, 60)	91	71 (59, 84)
Health Canada ^3^	I-Very High Importance	97	88 (73, 107)	98	95 (78, 115)	88	42 (33, 53)	95	51 (39, 68)
	II-High Importance	98	221 (194, 252)	100	248 (218, 283)	99	114 (95, 138)	100	141 (122, 165)
	III-Medium Importance	60	17 (13, 23)	87	65 (45, 94)	75	25 (18, 35)	69	12 (8, 17)
	IV-Low Importance	92	89 (76, 105)	94	113 (99, 130)	73	48 (39, 60)	91	71 (59, 84)
Total		99	416 (362, 477)	100	520 (455, 595))	100	229 (189, 278)	100	275 (232, 328)

^1^ Oral other than in the feed. ^2^ World Health Organization’s ranking of medically important antimicrobials according to the 6th revision [27]. Note that no veterinary products for cattle are marketed in Canada in the category “Important antimicrobials”. ^3^ Health Canada’s categorization of antimicrobial drugs based on importance in human medicine [28].

**Table 2 microorganisms-09-01106-t002:** Concordance correlation coefficients (CCC) with their 95% confidence interval (95% CI), and mean differences with their 95% CI, calculated from 101 or 97 dairy farms from Québec, Canada, to report agreement between a garbage can audit (REF method) and veterinary invoices (VET method), a governmental database (GOV method), and farm treatment records (FARM method), for quantification of usage rate of antimicrobial agents from products other than medicated feed (in DCDbovCA/100 cow-years) by route of administration, by category according to the World Health Organization (WHO) or to Health Canada, and in general.

Antimicrobial Category		REF vs. VET (101 Farms)		REF vs. GOV (101 Farms)		REF vs. FARM (97 Farms)	
		CCC (95% CI)	Mean Difference (95% CI)	CCC (95% CI)	Mean Difference (95% CI)	CCC (95% CI)	Mean Difference (95% CI)
Route of administration	Intramammary for lactating cows	0.95 (0.93, 0.97)	18 (−154, 190)	0.49 (0.38, 0.59)	−141 (−538, 256)	0.50 (0.37, 0.62)	−123 (−553, 307)
	Intramammary at dry-off	0.86 (0.80, 0.90)	0 (−47, 47)	0.20 (0.09, 0.31)	−48 (−140, 44)	0.74 (0.63, 0.81)	−10 (−71, 51)
	Injectable	0.90 (0.85, 0.93)	13 (−40, 65)	0.64 (0.53, 0.73)	−18 (−97, 61)	0.68 (0.58, 0.77)	−26 (−101, 49)
	Oral ^1^	0.11 (0.03, 0.20)	59 (−173, 291)	0.17 (−0.01, 0.34)	10 (−105, 125)	0.39 (0.21, 0.54)	7 (−77, 92)
	Intrauterine	0.28 (0.19, 0.36)	14 (−31, 59)	0.26 (0.14, 0.37)	10 (−23, 43)	0.14 (−0.02, 0.29)	6 (−26, 38)
	Topical	-	1 (−7, 8)	-	0 (−5, 6)	-	0 (0, 0)
WHO ^2^	Highest Priority Critically important	0.96 (0.94, 0.97)	7 (−49, 63)	0.47 (0.36, 0.56)	−49 (−186, 87)	0.55 (0.42, 0.66)	−42 (−186, 102)
	High Priority Critically Important	0.93 (0.90, 0.95)	6 (−44, 56)	0.52 (0.41, 0.62)	−30 (−125, 65)	0.51 (0.37, 0.62)	−27 (−132, 78)
	Highly important	0.50 (0.38, 0.60)	67 (−190, 325)	0.43 (0.29, 0.55)	−66 (−251, 120)	0.60 (0.48, 0.70)	−56 (−201, 88)
	Not used in humans	0.82 (0.75, 0.87)	24 (−57, 105)	0.37 (0.23, 0.50)	−41 (−173, 91)	0.47 (0.30, 0.60)	−20 (−157, 117)
Health Canada ^3^	I-Very High Importance	0.96 (0.94, 0.97)	6 (−46, 59)	0.46 (0.35, 0.56)	−46 (−176, 83)	0.62 (0.50, 0.72)	−38 (−164, 87)
	II-High Importance	0.92 (0.89, 0.94)	27 (−89, 144)	0.44 (0.32, 0.55)	−106 (−345, 132)	0.49 (0.36, 0.61)	−83 (−329, 164)
	III-Medium Importance	0.09 (0.03, 0.16)	47 (−186, 279)	0.22 (0.06, 0.37)	8 (−80, 95)	0.42 (0.25, 0.56)	−5 (−53, 43)
	IV-Low Importance	0.82 (0.75, 0.87)	24 (−57, 105)	0.37 (0.23, 0.50)	−41 (−173, 91)	0.47 (0.30, 0.60)	−20 (−157, 117)
Total		0.83 (0.76, 0.87)	105 (−237, 446)	0.44 (0.31, 0.55)	−187 (−673, 300)	0.51 (0.38, 0.63)	−145 (−635, 344)

^1^ Oral other than in the feed. ^2^ World Health Organization’s ranking of medically important antimicrobials according to the 6th revision [27]. Note that no veterinary products for cattle are marketed in Canada in the category “Important antimicrobials”. ^3^ Health Canada’s categorization of antimicrobial drugs based on importance in human medicine [28].

**Table 3 microorganisms-09-01106-t003:** Estimates of the total usage rate of antimicrobial agents (from products other than medicated feed) in dairy farms from Québec, Canada, according to veterinary invoices (VET method), the governmental database (GOV method), or farm treatment records (FARM method) when a given total usage rate of 0, 50, 100, 400, or 800 DCDbovCA/100 cow-years is reported using a garbage can audit (REF method). Estimates were calculated using negative binomial regression models applied to 101 (VET and GOV methods) or 97 (FARM method) dairy farms with the total usage rate of antimicrobial agents according to the REF method as the only predictor in each model.

Total AMU ^1^ Rate According to Method REF.	Estimated Total AMU ^1^ Rate According to Method VET	Estimated Total AMU ^1^ Rate According to Method GOV	Estimated Total AMU ^1^ Rate According to Method FARM
0	217	108	129
50	237	116	139
100	259	126	150
400	445	204	235
800	914	387	428

^1^ AMU rate: antimicrobial usage rate (in DCDbovCA/100 cow-years).

## Data Availability

The data presented in this study are available on request from the corresponding author.

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
