# Peer review of "Comparison of Quantification Methods to Estimate Farm-Level Usage of Antimicrobials Other than in Medicated Feed in Dairy Farms from Québec, Canada"

_microorganisms, 2021, doi:10.3390/microorganisms9051106_

Round 1

Reviewer 1 Report

A very interesting and straightforwardly executed study. The main question I have for the authors is:

How were non-bovine administered products (e.g., feline, canine, equine) separated out of the garbage can component relative to how such data was excluded from the software reported data for the analyses?

And some suggestions:

In the Conclusion, reiterate as was done in the Introduction why it is important to monitor AMU use (e.g., resistance) and perhaps reiterating also the distinction from medicated feed.

I think it could strengthen the paper to very briefly identify applications of this approach, e.g., if there was ever interest in monitoring AMU in equines or farm felines/dogs. Some readers of this paper might see benefits in being able to cross-reference between treatment records vs garbage can vs software to examine factors such as possible extra-label usage, or for an assessment of safe disposal methods or simply disposal practices. My point here is just with a small hint at other prospective applications, the authors can enhance the value of their work to the current application and give ideas for others.

Very nice work!

Reviewer 2 Report

The authors compared three quantification methods to the reference method (REF, garbage can audit) for monitoring antimicrobial usage from products other than medicated feed over one year in 101 dairy farms. The manuscript was well written, data was analyzed and presented in an appropriate way. There are some minor concerns should be addressed:

  1. The Introduction section should be shortened. The background of quantification methods for monitoring the surveillance of AMU in dairy farms in Quebec and Canada can partly move to the discussion section.
  2. If possible, the table with basic information of the farms included in this study may include in the manuscript, such as the animal scale, the location and the dairy production.
  3. The reference article or government document or website link should be provided in some text, such as “Since December 2018, a veteri-nary prescription is also needed to purchase all medically important antimicrobials in the other Canadian provinces” in the “Introduction” section. Please check all through the main text.

Reviewer 3 Report

Dear authors, I have read with interest your study. It looks promising and I think that some changes will improve it.

The abstract should be rewritten in order to present more of your findings. The methodology should be condensed.

Introduction - you need to add more references. 

Please view the pdf attached.

Round 2

Reviewer 3 Report

Dear authors, the article Comparison of quantification methods to estimate farm-level usage of antimicrobials other than in medicated feed in dairy farms from Québec, Canada present a lot of good improvements and it better present your research.